# Flexibility of functional neuronal assemblies supports human memory

Gray Umbach[1], Ryan Tan[2], Joshua Jacobs [3], Brad E. Pfeiffer [4] & Bradley Lega [2]✉

Episodic memories, or consciously accessible memories of unique events, represent a key aspect of human cognition. Evidence from rodent models suggests that the neural representation of these complex memories requires cooperative firing of groups of neurons on short time scales, organized by gamma oscillations. These co-firing groups, termed "neuronal assemblies," represent a fundamental neurophysiological unit supporting memory. Using microelectrode data from neurosurgical patients, we identify neuronal assemblies in the human MTL and show that they exhibit consistent organization in their firing pattern based on gamma phase information. We connect these properties to memory performance across recording sessions. Finally, we describe how human neuronal assemblies flexibly adjust over longer time scales. Our findings provide key evidence linking assemblies to human episodic memory for the first time.

Cognitive behavior, including memory formation and retrieval, requires cooperative but distributed neuronal firing[1,2], of which the self-organization of hippocampal neurons represents a key example. Data from animal models of hippocampal-dependent memory indicate that organized firing occurs on time scales of roughly 25 ms[3], a scale which has several theoretical advantages including facilitating spike-timing-dependent plasticity[4]. Subsequent work further demonstrated the importance of neuronal co-activity on these time scales[5–7] by identifying[5–12] and even incepting[9] memories represented by groups of co-firing neurons. Such data support the influential proposal that coordination of firing over longer time scales requires, at its root, the integration of multiple assemblies organized by gamma oscillations[1].

Further, groups of neurons that co-fire on a ~25 ms timescale ("gamma assemblies") may be specifically important for episodic memory formation related to the representation of context in the mesial temporal lobe (MTL). Recent reports have highlighted slow changes in the neural populations representing time and space ("drift") over time scales of minutes to days[6,13,14] and the coexistence of static and drifting spatial codes in the hippocampus[8]. These findings, along with computational modeling, indicate that assembly drift may reflect flexible memory updating and, indeed, that flexibility of assembly

formation is required for the accurate representation of temporal information[15]. Yet, evidence supporting the existence of gamma scale assemblies in humans remains limited. Previous work, including both cortical and MTL recordings, has identified neuronal firing sequences related to both item[16,17] and contextual[18] information on the order of hundreds of milliseconds[17], seconds[18], and minutes[16] but not on the fundamental gamma timescale proposed as the basic representational unit[1,19]. Co-firing patterns on the scale of milliseconds have been reported in the Rolandic cortex, though these were not linked to behavior[20].

Despite the importance of assembly formation on gamma time scales to episodic processes, there exists no data from MTL recordings to support their existence or mnemonic relevance in humans, including the mechanisms underlying their organization and the temporal dynamics of their activity. To address these key questions, we utilized a dataset of single-unit recordings in human epilepsy patients performing an episodic memory task[18]. We employ established methods to identify gamma timescale assemblies during episodic memory processing[5–7,21]. We then relate the spatiotemporal characteristics of human gamma scale assemblies, including drift in assembly activation strength, to memory behavior. Our findings reveal several new insights

[1]Department of Neurological Surgery, University of California San Francisco, San Francisco, CA 94143, USA. [2]Department of Neurological Surgery, University of Texas Southwestern, Dallas, TX 75235, USA. [3]Department of Biomedical Engineering, Columbia University, New York, NY 10027, USA. [4]Department of Neuroscience, University of Texas Southwestern, Dallas, TX 75235, USA. ✉e-mail: Bradley.Lega@UTSouthwestern.edu

into the characteristics of these fundamental electrophysiological components of mnemonic processing.

## Results

### Identification of mnemonic assemblies

Our study included 26 participants undergoing extraoperative seizure mapping coupled with microelectrode recordings from the MTL. Participants completed a total of 38 recording sessions. For each session, patients performed the free recall task, a common assay of episodic memory in which participants study wordlists of non-repeating nouns ("encoding") and are asked to recall ("retrieval") as many words as possible from the immediately previous list without cues[18] (Fig. 1A). Included participants recalled an average of 21.4 ± 9.1% (mean ± standard deviation) of the studied words (Fig. 1H). During task performance, we obtained microelectrode recordings from mesial temporal regions, including the hippocampus, entorhinal cortex, amygdala, and parahippocampal gyrus (Fig. 1B–D and Supplementary Fig. 1A). Based on extensive findings in rodent models[1,3,5–7], we focused our initial analysis on co-firing neurons occurring on a timescale of 25 ms, implementing previously published methods[21] using decomposition of neuron-by-25 ms time firing rate matrices. This method identifies assemblies as combinations of neurons whose spiking activity is sufficiently correlated within the specified timescale to conclude that the neurons are functionally codependent. Unlike other approaches that group cells on the basis of increased activity in response to the same stimulus[10,12], this framework selects for self-organizing patterns that persist across different items and context[1,3]. To achieve this, we first binned spiking data from all encoding events

into 25 ms non-overlapping time bins, calculating a normalized firing rate for each neuron in each bin. With a combination of principal and independent component analysis, we identified patterns of neural activity stemming from the co-linearity of cell firing. The ultimate result is a vector of assembly "weights" for each gamma assembly, with a single value for each recorded neuron (Fig. 1F). These weight values communicate the degree to which each neuron contributes to the assembly and are used to determine the expression of the assembly within each 25 ms time bin (Fig. 1E and Supplementary Fig. 2).

Using data from all encoding events across the recording sessions, we isolated a total of 45 gamma timescale assemblies in 15 of the recording sessions from 13 unique subjects (Supplementary Table 1, Fig. 1F, and Supplementary Fig. 1K). This significantly exceeded the count expected by chance, as determined by comparing the actual count to a null distribution generated by shuffling the spike trains before re-isolating co-firing patterns ($p < 0.001$ and 3.81 standard deviations above the distribution mean, Fig. 1G and Supplementary Fig. 1H). We only considered the 15 sessions with assemblies in all future analyses. We posit that we did not identify assemblies in the discarded sessions due to low single unit yield (Methods, Supplementary Fig. 1J). From the 15 sessions with identified assemblies, we isolated a total of 203 neurons for which we calculated common spike quality metrics which closely match those in previous studies of the human MTL[18,22] (Methods, Supplementary Fig. 1B–G). The number of simultaneously recorded neurons in these sessions ranged from 5 to 34, with an average of 13.5 cells. We defined assembly activation events (Methods) as 25 ms time bins during which assembly expression (Fig. 1E) exceeded the 95th percentile of values observed across the

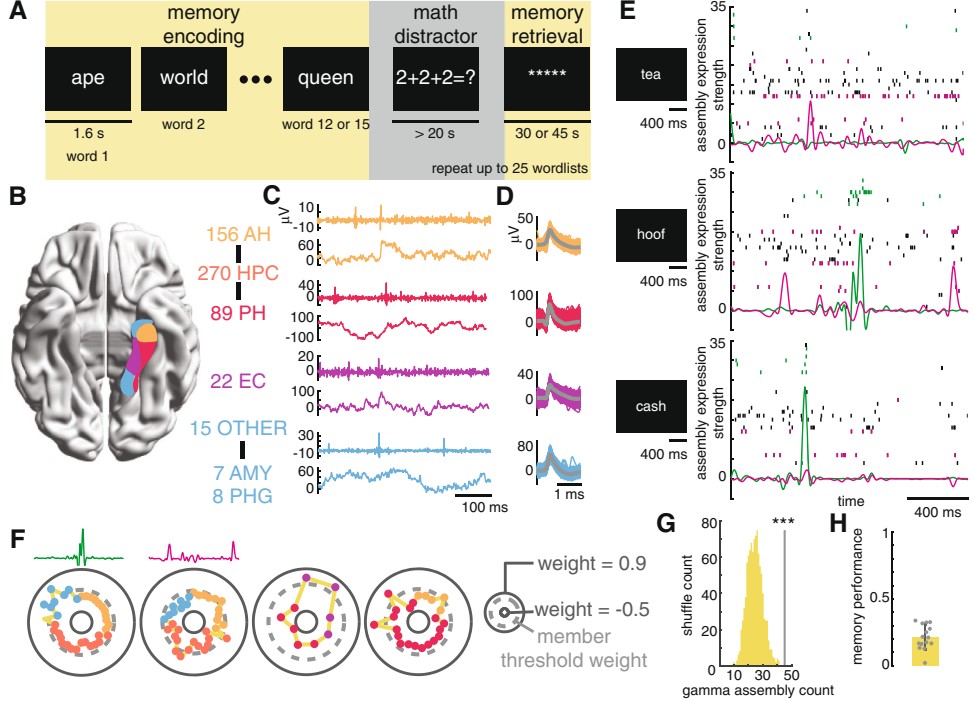

**Fig. 1 | Significant neuronal assembly identification during an episodic memory task. A** Schematic of the free recall task. Each displayed word the participant studies represents an "encoding event." **B** Unit yield for each brain region included in the study. **C** Example denoised high-frequency signals from which we isolated unit activity (top rows) and local field potentials (bottom rows) for each brain region. Coloring follows the convention shown in (**B**). Numbers next to the region indicate the number of units isolated from each region and include neurons from all sessions ($n = 307$), not only those from which assemblies were identified ($n = 203$). **D** Example units from each of the high-frequency signals displayed in (**C**). **E** Expression strength of two example assemblies superimposed on the pertinent spike rasters for three example encoding events. Expression strength curves and

spike rasters are colored to link them with the assemblies in (**F**). **F** Schematics of four example neuronal assemblies. The first two correspond to the data shown in (**E**). Each colored data point represents a neuron from that recording session. The further the data point from the circle's center, the greater the contribution of that neuron to the assembly, with member neurons falling outside of the dashed-line circle. The color of each data point represents the region of the neuron, as outlined in (**B**). **G** Comparison of the number of assemblies identified against a null distribution obtained by shuffling the spike trains (permutation test, $n = 1000$ shuffles, $p = 0.0009$). **H** Average and individual recall fraction of recording sessions with identified assemblies ($n = 15$). ***$p < 0.001$. Source data are provided as a source data file.

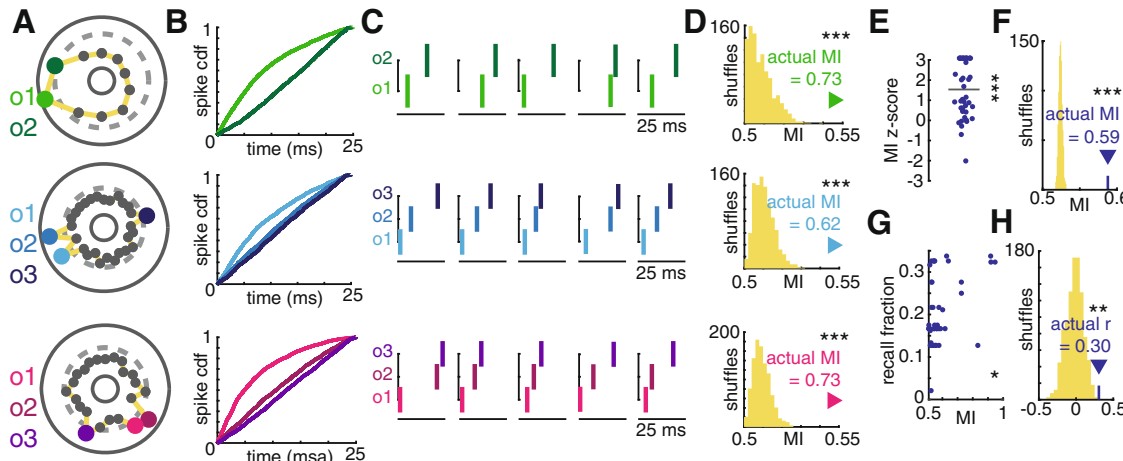

**Fig. 2 | Consistent firing sequences during assembly activation predicts successful memory and may result from member neurons locking to different phases of the underlying gamma oscillation. A** Schematics of three example assemblies with member neurons colored according to the expected firing order within activation events. **B** Cumulative distribution functions of spike times within activation events for each assembly member neuron, colored according to the key displayed in (**A**). **C** Spike rasters of assembly member neurons within five example activation events demonstrating the expected firing sequence. **D** Comparison of MI for each of the three assemblies from (**A**) to a null distribution obtained by shuffling spike times within activation events ($n = 1000$ shuffles, $p = 0.0009$). **E** Distribution of $z$-scores obtained by comparing the actual MI to the null distribution for each assembly (one-sample $t$-test, t($n = 45$) = 7.454, $p = 1.24$e-6). **F** Comparison of the average MI across assemblies to that of each shuffle (permutation test, $n = 1000$ shuffles, $p = 0.0009$). **G** Positive Spearman correlation between MI and overall recall fraction (spearman rank correlation, $n = 45$, $r = 0.305$, $p = 0.0419$). **H** Comparison of the observed MI-recall fraction correlation to a null distribution obtained by shuffling spike times before re-calculating (permutation test, $n = 1000$ shuffles, $p = 0.0020$). *$p < 0.05$. **$p < 0.01$. ***$p < 0.001$. Source data are provided as a source data file.

entire recording session. We observed more frequent assembly activation events following the presentation of recalled memory items ($t$-test, t(44) = 2.1, $p = 0.022$, permutation test, $p = 0.091$, Supplementary Fig. 3A–C, and S3E, Supplementary Fig. 4B). The degree to which assembly activity predicted successful memory encoding was strongly modulated by assembly complexity[5], a metric of how many neurons substantially contribute to assembly expression compared to how many neurons are available to contribute (see methods, Spearman rank correlation, $r = 0.61$, $p < 0.001$, permutation test, $p < 0.001$, Supplementary Figs. 3D, F, G, 4C). Neither assemblies (3 of 45, $p = 0.39$, binomial test) nor their constituent neurons (4/84 assembly member neurons versus 7/119 non-member neurons, $\chi^2 = 0.13$, $p = 0.71$, chi-square test) were significantly modulated by the semantic information of the words presented (Methods).

We applied these same methods to detect assembly activation on other time scales as well, reflecting organization at other gamma and theta frequencies. This analysis identified a maximum in the number and significance of assemblies at 25 ms, supporting the inferences drawn from rodent models that motivated our analysis focusing on this timescale[3]. We note that, as observed in rodent models[3], significant co-firing also occurs at other time scales (Supplementary Fig. 5), a point we address below in our Discussion.

## Assemblies are comprised of consistent neuronal firing sequences

To characterize the order of firing for assembly neurons within activation events, we adapted existing methods to relate consistent firing sequences within an assembly to memory behavior[23]. First, for each assembly, we isolated all spiking activity of member neurons ($n = 84$), defined as neurons that increased their firing rate above baseline and significantly more than non-member neurons (Supplementary Fig. 1L), across all activation events. We then defined a metric, match index (MI), that communicates how consistently the observed spiking data matches a possible template sequence based on ordered pairs of spikes (Methods). We tested the observed data against all possible orderings of the assembly member neurons and selected the sequence that maximized MI as the expected firing order (Methods, Fig. 2A–C

and Supplementary Fig. 6F–H). For example, the bottom two rows of Fig. 2 display data from assemblies with three member neurons. There are six orderings of a set of three cells. Within each activation event, we generate a list of ordered pairs from the assembly member neurons that fire. A firing sequence of member neuron 3-1-2, generated the ordered pair set of 3-1, 3-2, 1-2. We then calculated the fraction of observed ordered pairs that match each of the possible member neuron orderings to select the expected sequence. We observed that assembly member firing during activation events demonstrated significantly higher MI than chance ($t$-test, t(44) = 7.4, $p < 0.001$, permutation testing, $p < 0.001$, Fig. 2D–F and Supplementary Fig. 6I), indicating the emergence of consistent temporally ordered sequences in gamma scale assemblies. Further, the MI of assemblies positively correlated with the recall fraction observed during the corresponding session (Spearman rank correlation, $r = 0.30$, $p = 0.042$, permutation testing, $p = 0.0020$, Fig. 2G, H), supporting the mnemonic relevance of firing order consistency. This effect remained after controlling for assembly firing rate preferences for successful encoding (Methods, partial Spearman rank correlation, $r = 0.30$, $p = 0.047$) and average assembly expression strength across the session (Methods, partial Spearman rank correlation, $r = 0.32$, $p = 0.033$), indicating that assembly firing order carries mnemonic information above assembly firing rate. We observed no correlation between assembly member neuron firing order and baseline firing rate (Spearman rank correlation, $r = -0.03$, $p = 1.0$, Supplementary Fig. 6J).

## Assembly firing sequences are organized by underlying gamma oscillations

Motivated by the theoretical[1,3,19] and observed[24,25] importance of gamma oscillations in organizing assemblies and spatial information respectively, we next evaluated the influence of gamma oscillations (40 Hz) on organizing the neuronal spiking into assemblies. During assembly activation events, we observed a peak in gamma oscillatory power at 40 Hz, along with a concomitant increase in power at 100 Hz, possibly reflecting ripple-like or multi-unit activity[26–28], supporting the presence of gamma oscillations during these events (Supplementary

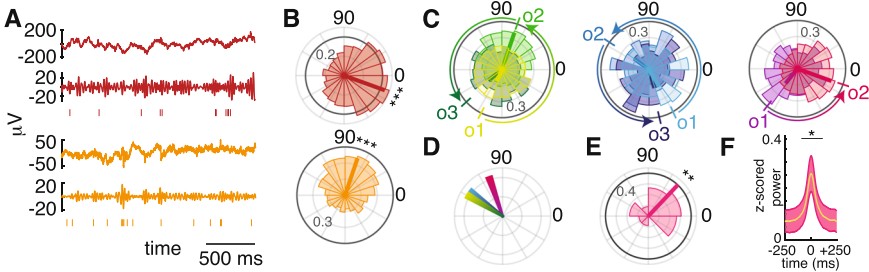

**Fig. 3 | Consistent firing sequences during assembly activation may result from member neurons locking to different phases of the underlying gamma oscillation. A** LFP (top), gamma frequency band activity (band-passed signal at 30–50 Hz, for display only), and spike raster for two example assembly member neurons demonstrating significant phase locking to the underlying gamma oscillation at 40 Hz. **B** Circular histograms demonstrating the probability density within each of the displayed phase bins for two examples of 40 Hz phase-locked assembly member neurons. Colors indicate that the neurons were isolated from the same recordings as the signal samples displayed in (**A**). The darkened line is at an angle equal to the median phase across all spikes and the length is equal to the mean resultant vector length. **C** Three examples of the relationship between firing order and preferred phase of spiking along the underlying 40 Hz oscillation. Phase histograms for each gamma phase-locked assembly member neuron for each assembly are superimposed. The median phase and mean resultant vector length are displayed for each neuron as in (**B**). Neurons firing later in the observed sequences (o2, o3) fire at gamma phases increasingly advanced from the phase of the primary neuron (o1) in the sequence. **D** Median pairwise phase offset from all neuron pairs of each assembly displayed in (**C**). Each median pairwise phase offset is colored according to the corresponding assembly from (**C**). **E** Demonstration of the non-uniformity of pairwise phase offsets across assemblies. The median pairwise phase offset and mean resultant vector length are displayed as in (**B**). **F** Significant gamma power occurs at the time of activation of gamma timescale assemblies. Black dots above a given time point reflects significant increases in 40 Hz power at that time after Bonferroni correction across time with an alpha of 0.05. The central line represents the mean z-scored power across assemblies and the surrounding shaded region represents the 95% confidence interval. \*$p < 0.05$. \*\*$p < 0.01$. \*\*\*$p < 0.001$. Source data are provided as a source data file.

Fig. 6A). To explore the possibility of this peak in the spectrum relating to ripples, we followed established methods[26] to identify ripple-like activity. Across recording sessions, we noted an average ripple frequency of $0.29 \pm 0.12$ Hz, comparable to the previous reports[26]. Assembly activations were 3.8 times more likely to occur within ripples than without, significantly higher than chance ($p = 0.005$, shuffle test, Supplementary Fig. 7D).

The 40 Hz gamma power change reflected a significant increase from baseline ($p < 0.05$ after Bonferroni correction across time, Fig. 3F). We assessed the influence of this oscillation on assemblies by measuring the phase locking of member neurons relative to this oscillation. Further, 75% of assembly member neurons (63 of 84, binomial test, $p < 0.001$) showed significant gamma phase locking (Rayleigh test, $p < 0.05$, Fig. 3A, B). Assembly member neurons were both more frequently ($\chi^2(1) = 5.25$, $p = 0.022$, Supplementary Fig. 5B) and strongly ($z(305) = 1.74$, $p = 0.041$, Supplementary Fig. 6C–E) phase-locked to 40 Hz than non-member neurons. Member neurons within the same assembly tended to lock to different phases of the underlying oscillation, with neurons firing later in the observed sequences (as defined by the firing order, see above) spiking later within the gamma cycle compared to neurons firing earlier in the observed sequences (Fig. 3C and Supplementary Fig. 6K). This is illustrated in Fig. 3E, which shows that the median phase difference between all pairs of gamma phased-locked member neurons was non-uniform with a positive (non-zero) phase offset (median phase difference = 47°, Rayleigh test, $z(29) = 4.7$, $p = 0.0082$, Fig. 3D, E).

### Assembly activation events are phase-locked to theta oscillations
We tested the theta phase at which assembly activation events occurred across the time series. For each assembly identified above, we measured the phase at which assembly activation events occurred relative to oscillations centered at 3, 5.5, and 8 Hz, predicated on the broad range of theta frequencies at which memory-relevant activity has been observed in humans. We used a Rayleigh test to identify significantly non-uniform phase distributions ($p < 0.05$, FDR-corrected), which revealed that 30 of 45 assemblies were significantly phase-locked ($p < 0.001$, binomial test). We then measured the average phase at which assemblies were found, using the theta frequency at

which maximum phase locking was observed (via the Rayleigh test described above). We found assembly activation events occurred at the trough of the theta oscillations, at all frequencies (Supplementary Fig. 7B). These observations suggest that the timing at which an assembly activates may be governed by theta oscillations, while the order of spiking within an assembly is modulated by low-frequency gamma oscillations.

### The flexible adjustment of neuron contribution to assemblies supports memory formation
Recent theoretical work posits that flexible assembly participation may be critical for episodic representation, especially "drift" of assembly activation over scales of minutes to hours[15]. This idea is also supported by experiments demonstrating dynamically changing neuronal codes for place[13] and time[14] in the hippocampus. Building on these ideas, we defined a metric, drift fraction (DF), that encapsulates the degree to which member neurons drift out of, and non-member neurons drift into, our observed assemblies. We calculated DF by looking for significant correlations between activation event number across the recording session and the firing rate of each neuron within those activation events (Methods). For example, if an assembly member neuron demonstrated a significantly decreasing firing rate across activation events (over the approximately 30-min recording session, top two rows of Fig. 4A), it was said to be drifting out of the assembly. The higher the DF, the greater the turnover in assembly member neurons (Fig. 4A, H and Supplementary Fig. 8A). We observed an overall DF of 21% across assemblies, higher than that observed in shuffle-generated null distributions ($p < 0.001$, Fig. 4B and Supplementary Fig. 8B). Interestingly, rather than these assembly alterations disrupting memory activity, assemblies with higher drift fraction correlated with greater memory recall performance (Spearman rank correlation, $r = 0.47$, $p = 0.0011$, permutation test, $p < 0.001$, Fig. 4C, D). To ensure this result did not stem from non-stationarity in our recordings (Methods), we compared the stability of assembly member and non-member neurons based on both spike waveform ($z(305) = -0.06$, $p = 0.95$) and firing rate ($z(305) = 1.69$, $p = 0.091$), observing no difference. Critically, the assembly drift fraction did not correlate with the proportion of non-stationary neurons recorded in the corresponding session ($r = -0.06$, $p = 0.69$, Supplementary Fig. 8C–F). Further, using sessions with sampling from both the anterior and posterior

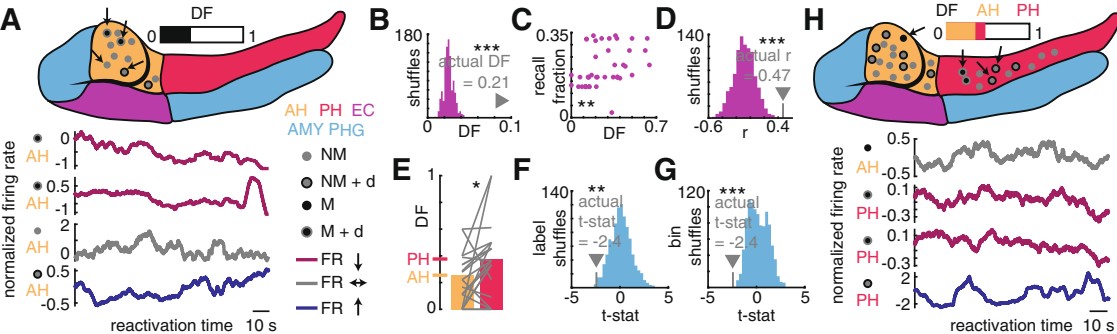

**Fig. 4 | Assembly drift predicts successful memory and occurs more frequently in the posterior compared to the anterior hippocampus. A** Top, schematic of the anterior (AH) and posterior (PH) hippocampus, entorhinal cortex (EC), amygdala (AMY), and parahippocampal gyrus (PHG) with all recorded neurons from an example session superimposed. The exact neuron location is for illustration only. Both member (M) and non-member (NM) neurons are displayed for an example assembly and neurons demonstrating drift across activation events (+d) are noted. The drift fraction is displayed above the schematic. Bottom, the firing rate of four example neurons from the recording session across all assembly activation events, showing examples of neurons that decrease (red) and increase (blue) their contribution and that remain stable (gray) across the session. For display purposes only, the firing rate curve is smoothed with a 10 s window and normalized and set the value at time zero to 0. The anatomical region and category of neurons corresponding to the displayed data were shown to the left of the plots. **B** Comparison of the observed drift fraction (DF) across all assemblies to a null distribution obtained by shuffling the activation bin order

for each neuron before re-classifying neurons (permutation test, $n = 1000$ shuffles, $p = 0.0009$). **C** Positive Spearman's rank correlation between the drift fraction and percentage of words recalled across the experiment ($n = 45$, $r = 0.470$, $p = 0.0011$). **D** Comparison of the observed DF-memory correlation to a null distribution obtained by shuffling activation bin order within each cell (permutation test, $n = 1000$ shuffles, $p = 0.0009$). **E** Pairwise comparison of DF between the AH and PH with the overall DF for each region displayed as the bar height and each line representing a single assembly (paired $t$-test, $t(n = 21) = -2.432$, $p = 0.0123$). **F** Comparison of the effect size of the observed pairwise DF difference along the hippocampal longitudinal axis to a null distribution generated by shuffling anatomical labels (permutation test, $n = 1000$ shuffles, $p = 0.007$). **G** The same as (**F**) but with the null distribution created by shuffling activation bins (permutation test, $n = 1000$ shuffles, $p = 0.0009$). **H** An example of asymmetric drift between the AH and PH, following the same convention as outlined in (**A**). However, AH and PH DFs are displayed separately above the schematic. Source data are provided as a source data file.

hippocampus, we observed clear differences in drift fraction along the hippocampal longitudinal axis, with greater turnover in the posterior hippocampus (anterior DF = 26%, posterior DF = 38%, $t(20) = -2.4$, $p = 0.012$, permutation tests following the shuffle of activation bin order, $p < 0.001$, permutation test following the shuffle of region label, $p = 0.0070$, Fig. 4E–H).

## Discussion

Neuronal assemblies have been extensively studied in rodents[3,5–12] as a critical functional unit of brain activity[1,2]. We identify such assemblies and correlate their activity with human episodic memory behavior, connecting assemblies with key aspects of human cognition. While we observe the preeminence of the 25 ms timescale among those we tested, as expected[1,3], we also identified significant assembly activity occurring at other time scales (Supplementary Fig. 5). Further work is needed to elucidate how different gamma timescale assemblies interact with each other with longer lasting neuronal co-activity, such as that defined by the period of a theta oscillation, the preferred information types of different assembly time scales (contrasting spatial and episodic information, for example)[1,19], and how gamma-timescale co-firing contributes to the organization of firing sequences observed in the human MTL occurring on the timescale of seconds to minutes[16–18]. We demonstrate that 25 ms assemblies reflect consistent sequences of single-neuron activity occurring within gamma oscillations (Fig. 2). The majority of assembly member neurons demonstrate gamma phase locking at 40 Hz, but to different phases (Fig. 3E), possibly giving rise to the observed firing sequences (Fig. 3C). In addition to the LFP power peak at 40 Hz during assembly activation, we identified a peak in the spectra around 100 Hz, possibly reflecting coincident high gamma or ripple activity seen during multi-unit activity[26–28]. While it is possible this peak relates to ripple-like activity, ripples in human data are considerably rare compared to rodent data and further dedicated work is needed to elucidate the recording devices (e.g., Behnke-Fried versus Utah array), frequency band, and prefiltering best suited to discern potential ripple from epileptic activity or physiologic high-frequency oscillations. Further, the free

recall paradigm imposes limited behavioral structure on item retrieval, which reduces our ability to observe ripple events as compared to spatial navigation paradigms. Regardless, interestingly, both the reliability of these sequences across task performance and the activation rate of assemblies positively correlates with successful memory (Fig. 2G and Supplementary Fig. 3C, D). Adopting the analogy employed by Buzsáki in describing the importance of gamma assemblies, our findings suggest that if assemblies can be thought of as the words the brain uses to build neural sentences that represent complex thought[1], then both the volume (activation) and spelling (activation order) of each individual word matters. Order of firing within an assembly ("letters in a word") may be subserved by the phase-locking of assembly neurons (Fig. 3C). Gamma phase-locking is known to organize place cell trajectories during sharp-wave ripple events[29] and the weight vector generated with our PCA/ICA assembly provides a hypothesis regarding the strength of the synaptic connections between neurons in the network[30]. However, we note that co-activity within gamma oscillations is not equivalent to strengthened physical connections by increased spine density thought to underlie assemblies[31], though the former may induce the latter via spike-timing-dependent plasticity[4]. Testing this in humans may be facilitated by ex vivo recordings of human tissue or pathological analysis of surgically resected tissue recorded in vivo[32]. We also observed complementary evidence that theta oscillations affect the timing of assembly activation (Supplementary Fig. 7), as compared to gamma oscillations that modulate the timing of spiking within an assembly. Our findings, therefore, connect human neuronal assemblies with important theoretical and empirical observations drawn from rodent models of episodic processing[33]. Examining these properties using paradigms that require ordered representations (such as serial recall) can further clarify the distinct roles of these mechanisms, analogous to place cell activity in rodent models of episodic memory.

Further characterization of the temporal dynamics of assembly formation can build on our findings related to assembly drift. We found evidence for both stable and drifting assemblies

(Supplementary Fig. 6B). However, participants with drifting assemblies demonstrated superior memory performance. While further work is needed to elucidate the mechanism and meaning of drift in memory, our finding is consistent with recent suggestions that dynamic assemblies may facilitate memory by allowing for flexible updating of experiential detail[15]. It is possible that the drift we observed, particularly in the posterior hippocampus, reflects our ability to update details of experiential context, increasing our ability to orthogonalize events within a broader experience, facilitating memory. The coexistence of both drifting and non-drifting assemblies is supported by theoretical[15] and empirical[8] work. Memory testing over long time scales (several days) could begin to probe how these classes of assemblies cooperate during human memory behavior. Interestingly, our finding that drift occurs more frequently in the posterior, compared to the anterior, hippocampus (Fig. 4E) offers a potential connection between theories of hippocampal longitudinal specialization related to "gist" versus "detailed" representations[34] and a proposed role for assembly drift to support memory updating[15]. It is possible that the anterior hippocampus maintains foundational features of an experience while the posterior hippocampus updates the immediately relevant representation as new events occur. This finding may have eventual implications for diseases that demonstrate selective abnormalities along the anterior-posterior hippocampal axis, such as post-traumatic stress disorder[34].

Given the experimental paradigm we used to identify assemblies ("single shot" encoding of non-repeated items), a natural question to ask is: what type of information do these assemblies represent[1]? Assembly activity occurs flexibly across the recording session, so they are not directly analogous to putative "engrams"[17], the activity of which can decode individual and specific memory items. Comparison of assemblies defined on gamma time scales and these broader representations may be facilitated by paradigms with a mix of repeated and non-repeated memory items. The assemblies we report exhibit clear associations with memory performance, and they demonstrate similarities to place cell and time cell population activity in terms of assembly drift. We did not see evidence of the assemblies responding to semantic information; however, future efforts utilizing experimental paradigms[35] suited to parse between semantic and temporal information could better elucidate this distinction. Therefore, it is possible that these assemblies represent the mnemonically relevant contextual features of the overall experimental experience, much like place and time cell ensembles, though we cannot exclude their representation of item-level information. While additional experimentation will be required to differentiate item and context representations at the assembly level, the fact that morphological (Supplementary Fig. 3D), anatomical (Fig. 4E), sequential (Fig. 2G), and temporal (Fig. 4C) features of the assemblies are all implicated in memory performance suggests assembly identification on gamma time scales represents an effective way to understand MTL activity during mnemonic representations.

The identification of mnemonically relevant human MTL assemblies harbors implications for neuromodulation strategies. Hampson et al[36]. describe the use of multi-input, multi-output state space models to provide a template for artificial recreation of spiking activity using intracranial electrodes. Our findings establish principles for identifying possible targets for neuromodulation using these approaches, as well the favorable time scales for co-firing and updating/modifying assemblies over the timescale of minutes to hours. Further, the timescale for assembly organization suggests an effective strategy for neuromodulation of memory may include the union of microstimulation using assembly organizational templates with macrostimulation designed to alter hippocampal gamma activity[37]. However, the importance of assembly identification may extend beyond mnemonic processing, including strategies for the amelioration of psychiatric illness[38].

## Methods

### Subjects and electrophysiology

We enrolled 26 human epilepsy patients who completed a total of 38 recording sessions for the study. We have previously described this population, and this work follows the same procedures[18]. All participants consented to a protocol approved by the institutional review boards at Thomas Jefferson University Hospital (TJUH) and the University of Texas Southwestern Medical Center (UTSW). Of the 26 human participants enrolled, 13, completing a total of 15 recording sessions, had an adequate single unit yield to identify assemblies and formed the focus of the study. Participants were implanted with Behnke-Fried electrodes allowing for identification of both local field potential (LFP) activity and single units, as previously described[18]. Neuroradiologists confirmed anatomical locations after co-registration of the pre-operative MRI with the post-operative CT[18]. We used BrainNet Viewer[39] to visualize electrode localizations.

Data preprocessing[40] and spike sorting[41] followed the same process as our previous report[18]. We only considered neurons with a baseline firing rate of at least 0.5 Hz. This resulted in a total of 307 neurons (hippocampus = 270, entorhinal cortex = 22, amygdala = 7, and parahippocampal gyrus = 8) (Fig. 1B), 203 of which came from sessions in which we isolated assemblies. We invariably isolated assemblies in recording sessions from which we isolated at least 13 units. For each neuron, we calculated spike quality metrics including a percentage of action potentials violating a 3 ms inter-spike interval (ISI), a ratio of the spike peak voltage to signal noise, number of units isolated per channel, isolation distances of units extracted from the same channel, average firing rate and burst index across recording[18] (Supplementary Fig. 1B–G). Note that while we cannot exclude the presence of multi-units in our recordings, our spike sorting algorithm endeavors to isolate single units[41] and we calculated the isolation distance between all neurons isolated on the same channel, finding values consistent with previously reported high-quality datasets[22].

For LFP analyses, we low-passed the signal at 300 Hz. We subsequently applied a notch filter at 60 Hz and downsampled by a factor of 30 (UTSW) or 32 (TJUH) to an approximate sampling rate of 1000 Hz. For gamma phase and power analyses, we used Morlet wavelet (with six cycles) convolution at 40 Hz as its oscillation period matched our selected assembly timescale (25 ms)[3] and we observed a peak in gamma power at this frequency coincident with assembly activity (Supplementary Fig. 6A). To remove the influence of spiking activity, we excised each spike event and bridged the two segments of data with linear interpolation.

While patients with hippocampal sclerosis commonly do not require intracranial monitoring, it is possible microscopic changes consistent with mesial temporal sclerosis may be present in some of our patients. Of note, patients' antiepileptic drugs are adjusted on the basis of clinical necessity throughout the monitoring period, rendering this influence on function impossible to assess. We did not include any sessions with observable seizure activity or aura. Further, the clinical team flagged excessively noisy channels[18]. These channels were discarded. Channels exhibiting inter-ictal activity at any point during implantation were also flagged. Though we did not exclude these channels from analysis, we ensured our ability to identify assemblies following their exclusion (Supplementary Fig. 1H).

### Task

The task design is the same as in the previous work[18]. In brief, participants studied up to 25 wordlists comprised of 12 (UTSW) or 15 (TJUH) monosyllabic nouns on a laptop computer at the bedside. Word stimuli were not repeated. Each word remained on screen for 1.6 s, followed by a jittered inter-stimulus period of 0.8–1.2 s. Patients then completed two-step arithmetic problems of the form A + B + C = ?, where A, B, and C were single-digit integers, for at least 20 s. Finally, they verbally recalled as many words as possible from the immediately preceding

wordlist without receiving any cues. They had 30 (UTSW) or 45 (TJUH) seconds for recall.

## Assembly identification

We isolated assemblies using a framework[21] previously utilized in rodent studies[5–7]. For each recording session, we calculated the firing rate of each neuron within non-overlapping time bins spanning the experiment. We tested various time scales corresponding to the periods of previously investigated gamma and theta frequency bands. We selected 25 ms as it is both the theoretical[1] and demonstrated[3] (Supplementary Fig. 5) optimal timescale of assembly organization. We then normalized (z-scored) the binned firing rates for each neuron to avoid biasing the result towards neurons with higher baseline firing rates. After normalizing, we constructed a neuron-by-time bin matrix. To isolate assemblies, we used all time bins falling within all encoding events across the recording session.

Next, we performed principal component analysis (PCA) on the normalized firing rate matrix, generating an eigenvalue associated with each identified principal component. According to the Marchenko–Pastur law, eigenvalues outside of the bounds:

$$\left[\left(1 - \sqrt{\frac{n}{b}}\right)^2, \left(1 + \sqrt{\frac{n}{b}}\right)^2\right] \tag{1}$$

where $n$ = the number of neurons and $b$ = the number of time bins, stem from a correlation between the variables (neurons) within the matrix exceeding that expected if the variables were independent. Therefore, any isolated pattern whose associated eigenvalue exceeds this upper bound is the result of significant neuronal co-firing. While it is possible to define cell assemblies on the basis of PCA patterns, previous work has demonstrated that stopping there leads to the mixing of the true cell assemblies[21]. Therefore, following the aforementioned PCA analysis, we performed an independent component analysis. To do so, we first projected the firing rate matrix onto the PCA patterns with eigenvalues exceeding the upper limit delineated by the Marchenko–Pastur law:

$$R_p = P_{co}^T R \tag{2}$$

where $R_p$ is the firing rate matrix projected onto the significant principal components, $P_{co}^T$ is the transpose of the matrix with the coefficients for each neuron across all significant principal components, and $R$ is the original normalized firing rate matrix. This, in effect, forces the ICA to return only as many assemblies as there were eigenvalues exceeding the upper bound expected for independently firing neurons. We utilized the MATLAB implementation of the fastICA algorithm[41] to solve for the unmixing matrix, $U$, using $R_p$. We then used the unmixing matrix to rotate the PCA-derived assemblies, given by $P_{co}$:

$$W = P_{co} U \tag{3}$$

where $W$ is a neuron-by-assembly matrix with each column containing the weight vector for the index assembly. The weight vector communicates the degree to which each neuron contributes to each assembly, with arbitrary scale and direction. We then transformed each weight vector into a unit vector, setting the sign such that the greatest assembly weight was positive as previously described[5].

While each neuron contributes to each assembly to some degree, we defined "assembly member neurons" as those with weight vector values exceeding the mean by at least one standard deviation. We then independently confirmed that this definition isolated neurons that were both significantly more active than baseline and more active than non-member neurons during activation events (Supplementary

Fig. 1L). We required an assembly have at least two member neurons to avoid assemblies dominated by a single recorded neuron.

We additionally calculated the complexity, referred to as sparsity in the previous work[5]:

$$1 - \frac{\sqrt{n} - \sum |w_i|}{\sqrt{n} - 1} \tag{4}$$

where $w_i$ is the weight vector value of the $i$th neuron, and $n$ is the length of the weight vector (equal to the number of recorded neurons in a given session). High complexity communicates that the assembly has a high proportion of recorded neurons contributing to the assembly.

## Assembly activation

We calculated the "expression strength," the degree to which an assembly is active, with the following equation:

$$E(b) = R(b)^T O_i R(b) \tag{5}$$

where $E(b)$ is the expression of assembly $i$ in the time bin of interest, $b$, $R(b)$ is the firing rate matrix column corresponding to time bin $b$, and $O_i$ is the outer product of the weight vector of assembly $i$. For each assembly, we calculated the expression strength within all time bins across the recording. We defined assembly "activation events" as time bins during which the expression strength exceeded the 95th percentile of all observed expression strengths for that assembly. Note that activation does not refer to memory retrieval.

## Assembly identification controls

The PCA/ICA assembly identification framework depends on the assumption that a matrix storing the firing rates across time of a group of independently firing neurons will not have an eigenvalue exceeding the Marchenko–Pastur limits[42]. However, matrices storing values of a parameter whose distribution is heavily skewed may result in eigenvalues that spuriously exceed these limits[43,44]. This is a theoretical risk in firing rate data as they are classically Poisson distributed. However, the risk of tempered by the size of the matrix used for PCA[43]. Further, eigenvalues beyond the expected bounds in these cases result from one or more instances of extreme values within the matrix. To reduce the risk of extreme values, we limited our analysis to neurons with a baseline firing rate of at least 0.5 Hz.

However, to confirm that the identified assemblies could not be explained by statistical error, we implemented a shuffle control. For this, we randomly shuffled the spike trains of all neurons before re-isolating assemblies. We then compared the actual number of assemblies to this null distribution, noting that the observed count exceeded the shuffle counts in every case (Fig. 1G).

We calculated a spatial distribution score to capture the degree to which assemblies clustered spatially (Supplementary Fig. 1I). To do so, for each assembly, we noted the channel, region, and hemisphere of each assembly member neuron. We then took all pairs of assembly member neurons. If the neuron pair shared the same channel, it received a score of 1. If it shared the same location (for example, both in the entorhinal cortex), it received a score of 2. If it shared the same hemisphere, it received a score of 3. If the neurons were from different hemispheres, it received a score of 4. We summed across all neuron pairs to obtain a single distribution score for each assembly, with lower scores representing more localized assemblies. We then randomly selected neurons from the same recording session, irrespective of assembly membership, to generate a null distribution of distribution scores. If an assembly had four neurons, we randomly selected four neurons for each iteration. We then compared the actual distribution score to the null distribution, generating a z-score for each assembly. We tested across assemblies against zero with a two-sided t-test to

assess if observed assemblies were more localized or distributed than expected by chance.

We additionally tested for spatial clustering separately for the hemisphere, region, and channel. We did so by counting the instances in which member neuron pairs had the same hemisphere, region, and channel. We then counted the number of times each neuron pair from all included recording sessions overlapped by hemisphere, region, and channel. We then performed a chi-square test for each variable separately to see if assembly membership modulated spatial overlap. Results are found in the figure legend of Supplementary Fig. 1.

We assayed the degree to which assemblies and their constituent neurons were modulated by semantic information by first performing latent semantic analyses on all words included across sessions of the experiment, following our previously established methods[18]. We calculated the cosine similarity between each word and a reference word, "world", to assign each stimulus a numerical score that represents its meaning. We then evenly split the words into ten groups based on this numerical score, with each word possessing more semantic similarity within groups than across groups. To determine if an assembly responded to semantic information, we performed a one-way ANOVA with assembly expression strength during encoding events (the time during which the word stimuli were displayed) as the dependent variable and semantic group as the independent variable. To assay each assembly member neuron's responsiveness to semantic information, we likewise performed a one-way ANOVA with neuron firing rate as the dependent variable.

### Intra-assembly neuronal sequences

To identify firing sequences within assembly activation events, we adapted a pair matching framework previously employed in the rodent place cell sequence replay literature[23]. We only used assembly member neurons for this analysis. In contrast to place cell replay where the expected firing sequence is known (the sequence observed during behavior), we did not have an expected firing sequence before searching for sequence development. Therefore, for each assembly, we generated a list of all possible orderings of neuron firing. For example, if an assembly had only two member neurons, the sequence set was simply [1-2, 2-1]. We then used each possible sequence as a template, measuring how well the observed firing sequences matched each template.

We looked within all activation events to compile a list of all observed ordered pairs of assembly neuron firing. For example, an assembly with three neurons which all fired in an activation event in the observed order 2-1-3 generated the ordered pair set of [2-1, 2-3, 1-3]. We calculated the fraction of ordered pairs that matched each template sequence (defined as "matching index", MI), selecting the template sequence with the expected sequence for that assembly. Rare instances of simultaneous assembly member spiking were not considered.

Due to this procedure, the minimum MI for an identified expected sequence was 0.5. Therefore, we performed permutation testing to define the statistical significance of an observed MI. We did this by repeating the aforementioned procedure 1000 times after shuffling the spike times within activation events each time. We then compared the actual MI to the null distribution of MI values to generate a z-score as follows:

$$z_{MI} = -\text{norminv}\left(\frac{\text{sum}(\text{MI}_s > \text{MI}_r) + 1}{\text{length}(\text{MI}_s) + 1}\right) \quad (6)$$

where $\text{MI}_s$ is the distribution of shuffled MI values and $\text{MI}_r$ is the MI value obtained from unshuffled data. We used the equation within the parenthesis to generate p values from shuffled data throughout the manuscript. We compared the distribution of z-scores, one for each assembly, to zero with a one-sided t-test (Fig. 2E). We also compared

the average MI across assemblies to the distribution of those seen in this null distribution (Fig. 2F).

We calculated the Spearman rank correlation between assembly MI values and the recall fraction from the corresponding session, the ratio of words recalled to words studied across experimentation. We compared the observed correlation coefficient to the distribution of coefficients obtained following the shuffling of the spike times within activation events, as described above. As a further control, we calculated a partial Spearman rank correlation between assembly MI values and the recall fraction, using each assembly's rate-coded subsequent memory effect (see below) and session-wide average expression strength, markers of assembly activity during successful encoding events and across the session, respectively, as the controlling variable.

### Assembly activation-triggered gamma power

We extracted the power at each of 75 linearly spaced frequencies ranging from 2 to 150 Hz for the 500 ms (−250 to +250 ms) surrounding each activation event, utilizing Morlet wavelets. After extracting raw power, we log-transformed and z-scored the values to allow for averaging across channels. For each assembly, we extracted z-scored power for each channel, recording at least one assembly neuron. We then averaged across channels and activation events to yield a single z-scored power time series for each assembly. We then performed a t-test across assemblies, comparing the averaged z-scored power across assemblies to zero (session-wide baseline power) for all time-frequency points. This procedure at 40 Hz generated the plot seen in Fig. 3, where we Bonferroni-corrected across time to identify significant deviations from baseline. To identify which frequencies were maximally coincident with assembly activity, we transformed the p value at each time-frequency point obtained by the t-test and normalized the power values observed in the 100 ms surrounding assembly activation (−50 to +50 ms) by the average power values observed at each frequency in the 400 ms surrounding this epoch (−250 to +250 ms). This procedure identified two peaks in baseline-normalized assembly activation-triggered power: 40 and 100 Hz (Supplementary Fig. 6A). We selected the former for further analysis due to its theoretical role in organizing 25 ms assemblies[1,3].

### Gamma phase-locking neurons

As we identified co-firing activity on the timescale of 25 ms, the period of a 40 Hz gamma oscillation, and noted a peak in the assembly activation-triggered power at 40 Hz (Supplementary Fig. 6A), we used Morlet wavelets at 40 Hz to extract gamma phase and power. Pre-processing of the LFP is described above. We extracted the gamma phase at each spike time for each assembly member neuron. Then, for each neuron, we tested the phase distributions for non-uniformity with a Rayleigh test. We called a member neuron gamma phase-locked if the p value from the Rayleigh test was below 0.05. We tested for an above chance proportion of gamma phase-locked neurons with a one-sided binomial test with a chance rate of 0.05.

Next, for each assembly, we identified all member neurons that demonstrated gamma phase locking. Then, for each neuron, we calculated the median phase of all spike phases observed during memory behavior. We then calculated the median pairwise phase difference between all member neurons for each assembly. We calculated the pairwise phase difference in such a way that positive differences communicated that neurons later in the expected firing sequence fired at a later phase. We tested the distribution of median pairwise phase differences for non-uniformity with a Rayleigh test (Fig. 3E).

We compared both the proportion and strength of gamma phase locking between assembly member and non-member populations. We compared the proportion of gamma phase locking neurons between groups with a chi-square test. We compared the strength of phase locking by assessing for a difference in the Rayleigh Z distributions of the two groups with a rank-sum test. Additionally, we controlled for

differences in firing rate between the two groups by first testing for a difference in baseline firing rate between them with a rank-sum test. Next, we randomly downsampled the phases contributing to the Rayleigh test for each neuron, such that all assembly members and non-members had numbers of contributing spikes. We then took the median of all obtained Rayleigh Z values for both groups and compared them. We repeated this process 200 times and assessed for a deviation in groupwise difference with a one-sample $t$-test against zero (Supplementary Fig. 6D).

### Theta phase-locking assemblies

Just as we tested for gamma phase locking of neurons, we tested for theta phase locking of the assembly activations. For each channel in each recording session, we extracted the theta phase at 3, 5.5, and 8 Hz during each assembly activation. We calculated phase locking using the Rayleigh test at each phase and channel, and then FDR-corrected across phase and channel ($Q = 0.05$). If a given assembly had at least one channel at one frequency that survived correction, we said that assembly demonstrated significant phase locking. We used a binomial test to determine if the proportion of assemblies that demonstrated phase locking significantly exceeded the chance.

Next, we compiled the mean phases of all assemblies demonstrating significant phase locking at 3, 5.5, and 8 Hz (23, 26, and 25, respectively). We then tested for non-uniformity of the distribution of mean phases with a Rayleigh test. These results are displayed in Supplementary Fig. 7.

### Ripple analysis

We followed established methods for the identification of ripple acitivity[26]. In short, we band passed the raw EEG from 80–120 Hz and then used a Hilbert transform to extract power within that band. We identified segments of the signal where (1) the ripple band power exceeded 2 standard deviations above the mean across the recording, (2) power exceeded this threshold for at least 25 ms, and (3) the maximal power exceeded 3 standard deviations above the mean across the recording. We assigned each ripple a start and an end time. To minimize the risk of misinterpreting pathologic activity as ripple activity, we first applied an automated artifact rejection algorithm previously used in the context of ripple identification in humans[26]. We first $z$-transformed the EEG time series and the first derivative of the EEG signal. We marked all times at which the $z$-score of the signal or the first derivative exceeds 5. We removed 100 ms before and after all such time points.

Following the identification of ripples, we calculated two metrics to determine the significance of their association with assembly activations. We calculated the % of assembly activations occurring during ripple activity (coincidence) and the ratio of the rate of assembly activations occurring within compared to outside of ripples (relative risk, RR). To assay significance, we randomly circularly shuffled the ripple times across the experiment and recalculated these metrics 200 times.

### Rate-coded subsequent memory effect

We calculated the activation rate during each encoding event by counting the number of activation events and dividing by the duration of the encoding event, 1.6 s. We then assessed for a subsequent memory effect (SME) for each assembly by comparing the activation rates during successful encoding events (SE) to those observed during unsuccessful encoding events (UE) with a one-sided rank-sum test. This generated a $z$-score communicating the degree to which assemblies increased their firing rate during SE compared to UE events. We tested for an effect at the assembly level with a one-sided $t$-test against zero. We additionally performed a Spearman rank correlation between the SME $z$-scores and the complexity scores for each assembly.

We controlled the observed SME effect and SME-complexity correlation by re-calculating SME $z$-scores 1000 times and shuffling the event labels each time. The shuffle procedure preserved the number of SE and UE events within the experiment but randomly reassigned the encoding events to each group.

### Assembly drift

We defined a metric, drift fraction (DF), to capture the degree to which the neurons contributing to assembly activation changed across recordings. For each assembly, we extracted the z-scored firing rates for all recorded neurons during each assembly activation event. We performed Spearman rank correlations between the $z$-scored firing rate and activation bin number for each neuron. Members with significant correlations ($p < 0.05$) and negative correlation coefficients were said to be "drifting out" of the assembly. Non-member neurons with significant positive correlations ($p < 0.05$, $r > 0$) were said to be drifting into the assembly. We defined the DF as the number of neurons drifting in and out of the assembly divided by the total number of neurons recorded.

We calculated the overall drift fraction by summing the numerators and denominators from all individual assembly DF calculations. We compared this value to a shuffle-generated null distribution to ameliorate the concern of bias in our correlation procedure. We generated this distribution by randomly shuffling the activation bin order before re-calculating the $z$-scored firing rate-activation bin correlation for each neuron (Fig. 4B). We next calculated the Spearman rank correlation between assembly DF and recall fraction. We compared the actual value to the distribution of correlation coefficients obtained by randomly shuffling the activation bins before repeating the procedure.

We investigated the possible influence of recording drift on our assembly identification and drift analysis. For this, we performed an augmented Dickey–Fuller test on spike waveform[18] and firing rate data for each assembly member and non-member neuron. For the spike waveform analysis, we first averaged the first 5 percent of all spikes observed during the experiment. We then calculated the Euclidian distance between this average waveform to all spikes observed throughout the remainder of the experiment. We used the vector of distance values for the augmented Dickey–Fuller test. For the firing rate analysis, we binned the spiking activity across the experiment into 100 equal duration time bins, calculating the firing rate for each bin. We then used this vector of firing rate values for the statistical procedure. We used both metrics to compare the stationarity of assembly member and non-member neurons with a rank-sum test. Further, we correlated the fraction of units in a session potentially non-stationary by firing rate criteria with the observed drift fractions of assemblies isolated from the corresponding sessions.

### Longitudinal differences in assembly drift

Following previous observation that the posterior hippocampus (PH) is more detail-oriented than the anterior hippocampus[34] (AH) and that drift may reflect memory updating[15], we compared the DF observed within the AH and PH. We did this by identifying all assemblies with sampling from both the AH and PH ($n = 21$ of 45). We then calculated the DF for the AH and PH separately. Next, we performed a paired one-sided $t$-test (Fig. 4E). We controlled this result by generating two shuffled-derived null distributions. For the first (Fig. 4F), with each session, we randomly shuffled the anatomical labels of neurons before re-calculating the AH and PH drift fractions. For the second (Fig. 4G), we randomly shuffled the activation event order for each assembly before re-calculating AH and PH drift fractions.

### Statistical testing

Unless otherwise specified, we used the Anderson–Darling test to assess for normality before using parametric tests. Otherwise, we used nonparametric statistics. We used an alpha of 0.05 unless otherwise

noted. All other relevant statistical information can be found located under the appropriate heading.

## Reporting summary

Further information on research design is available in the Nature Research Reporting Summary linked to this article.

## Data availability

Source data are provided with this paper.

## Code availability

The EEG Toolbox, a suite of MATLAB functions, is available at https://memory.psych.upenn.edu/Software. The custom code used for this paper is available upon request to the corresponding author (B.L.).

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

## Acknowledgements

We thank Michael D. Rugg for the helpful remarks on the manuscript. We are thankful for each patient who participated in the study. The project was supported by NIH R01NS107357 and R01NS125250 to B.L., R01-MH104606 to J.J., and the Southwestern Medical Foundation to B.E.P.

## Author contributions

G.U. and B.L. conceptualized the study. G.U., J.J., B.E.P., and B.L. developed the methodology. R.T. and B.L. performed the investigation. G.U. generated data visualization. J.J., B.E.P., and B.L. acquired funding. B.L. administered and supervised the project. G.U. and B.L. wrote the original draft. G.U., J.J., B.E.P., and B.L. reviewed and edited the draft.

## Competing interests

The authors declare no competing interests.
