## [Peer Review File · Nature Communications]

Flexibility of Functional Neuronal Assemblies Supports Human MemoryREVIEWER COMMENTS

Reviewer #1 (Remarks to the Author):

In the paper entitled “Flexibility of functional assemblies supports human memory”, the authors used data from refractory epileptic patients to investigate the role of gamma activity in different brain structures during memory task. The paper is very interesting. Main novelty resides in using advanced methods for unit activity inherited from animal research in human subjects performing a memory task. I find the paper promising and well written in terms of general narrative, which flows logically from premises to conclusions. Method section however, lacks some important details to make it more understandable and, more importantly, reproducible. For instance, the incredible number of different approaches used to identify and characterize neuronal assemblies would benefit from a graphical representation linking raw data to raster to assemblies, and finally to statistics. Moreover, figures are very blurred in my document which makes very difficult to read some important details such axis information. Things get worst if one tries to zoom in to look at details (e.g. axis labels). That’s annoying also because they all look very dense of well-presented results but difficult to carefully revise.

I have several concerns listed below:

Data description is very confusing across the entire MS. Numbers keep changing across different paragraphs without clear connection with the unique independent variable that is meaningful which are the different N-subjects. For instance, authors says that the initially enrolled N=26 epileptic patient for a total of 38 recording sessions. How many recording sessions per patients have been performed and which was the criteria for performing in a subset of patients more than one session? Later in the MS, authors states “We isolated a total of 45 gamma timescale assemblies in 15 of the recording sessions” from how many different subjects?

In respect to the task, subjects should recall as many words as they can from a list presented during encoding and after a distractor task (math). The authors said that those are non-repeating words, but are those words also non-semantically related and do the synonyms have been removed? If not, does the assembly characteristics predict encoding of semantically related words (or synonyms) better/worst?

There is no discussion about the potential role of pathology and/or about AEDs. According to data description, these patients are affected by refractory epilepsy undergoing presurgical evaluation for surgical resection. As far as I know, hippocampal sclerosis is a typical structural malformation in MTL epilepsy. The authors rejected pathological activity (although description of this particular step remains a little confusing, see below comment). Finally, anti-epileptic drugs are known to have mid to large

effects on cognitive performances including memory and motor action. These effects are strongly depending on the type of AED administered and duration of treatment. The authors should at least comment the potential role in performance linked to pathology and AEDs.

Method section, line 30: authors state that they flagged and discarded channels with inter-ictal activity while in the following sentence saying that they did not. That's a contradiction that should be fixed.

Minor:

Fig1 what the numbers next to label mean - e.g. 156 - AH?

Reviewer #2 (Remarks to the Author):

Umbach et al. studied hippocampal neural firing pattern in relation to gamma phase in a free recall task, using human cell data. They showed 25 ms assemblies reflect consistent sequences within gamma oscillations and phase locking.

The data is rare and interesting. But the function wrt. gamma oscillation remains elusive. For example, what is the function of phase locking, does it relate to autoassociative dynamics? e.g., Pfeiffer, B. E., & Foster, D. J. (2015). Autoassociative dynamics in the generation of sequences of hippocampal place cells. *Science*, 349(6244), 180-183.

Similarly, the function of "drift" remains unclear. Throughout paper, there are many metrics being computed but without clear reference to function.

Also, I don't think this the first time neural spiking in temporal lobe is shown to be related to episodic memory. "Neural assemblies" are also more like a analysis product (especially given there are only 13, simultaneously recorded neurons, on average). It is not clear why this is definitively better or revealing things fundamentally different compared to previous findings, e.g., Vaz, A. P., Inati, S. K., Brunel, N., & Zaghoul, K. A. (2019). Coupled ripple oscillations between the medial temporal lobe and neocortex retrieve human memory. *Science*, 363(6430), 975-978.

There are also some seems-too-strong inferences in the discussion with little empirical evidence, e.g., on page 6, line 210-212 "However, participants with drifting assemblies demonstrated superior memory performance, supporting recent suggestions that dynamic assemblies may facilitate memory by allowing for flexible updating of experiential detail", I don't see a clear logic here.

Minor:

Page 4, line 126-130, the author stated “This effect remained after including assembly firing rate preferences for successful encoding, ... , indicating that assembly firing order carries mnemonic information above assembly firing rate.” Do authors meant to write “excluding”?

Reviewer #3 (Remarks to the Author):

The authors investigated single-unit recordings in epilepsy patients performing a word memorization task with free recall. They describe the contribution of co-firing groups of neurons and local field potentials in the medial temporal lobe to memory formation. Their findings suggest that neuronal assemblies exhibiting correlated firing on a 25 ms timescale and spike-field coupling in the gamma range play an important role in successful long-term memory.

This investigation is based on a precious data set, the analytic approaches appear to be inventive and sound, and the results are interesting and relevant. In some instances, the interpretation of the results is hard to follow, from my view. Overall, I found that the study is analytically and mathematically very strong, but that there are several conceptual weaknesses and confusions. Moreover, from my view important information is missing in the manuscript. Please find enclosed more detailed comments and suggestion. Major comments are marked with underlined page numbers.

- Page 2: “Our study included 26 participants ...”. Please add here how many subjects/sessions entered the main analyses.
- Page 3: “To achieve this, we first binned ... into 25 ms time bins”. Please mention that the time bins were non-overlapping.
- Page 3: “We isolated a total of 45 gamma timescale assemblies ...”. Please add the information here that this procedure was based on data from the whole recordings sessions.
- Page 3: “We isolated a total of 203 neurons ...”. Did you differentiate between single- and multi-units?
- Page 3: “... quality metrics which closely match those in previous studies of the human MTL18”. Please also cite studies from other groups.
- Page 3: “We defined assembly reactivation events ...”. The term “reactivation” is misleading in this context. This term would only fit when assemblies were identified in an initial time interval and would be active in a later interval. The authors should, for instance, just speak of assembly activation, as well as of a subsequent memory effect.
- Page 3: “... we isolated all spiking activity of member neurons ...”. From my view, also the term “member neuron” is potentially confusing. Based on assembly identification, it would be intelligible to call the total of 203 neurons “member neurons”. The subset of neurons with increased firing rates during activation events could, for instance, be called “responsive” assembly neurons.

- Page 4: "... along with concomitant in power at 100 Hz ...": please elaborate on this finding in the discussion.
- Page 5: "... assemblies with higher drift fraction correlated with greater memory recall"; page 6: "... participants with drifting assemblies ...". In my opinion, it is unclear whether these findings indeed support the concept of drifting assemblies. Previous single-unit findings suggest that different words are represented by different groups of neurons, which however may overlap. Due to the overlap neurons can be identified which show a category-related response to visually presented words. These word-responsive neurons may contribute to the responsive assembly neurons identified by the authors. I would suggest that the authors check this by identifying word-responsive neurons and testing the overlap with the responsive assembly neurons. Importantly, the observed drift, in my opinion, may be just a reflection of the fact that different, however overlapping, assemblies represent different words. Hence, I further suggest that the authors reconsider their interpretation of these findings.
- Page 6: "The assemblies we report exhibit clear associations with memory performance ...". Therefore, it is possible that these assemblies represent the mnemonically relevant contextual features of the overall experimental experience ...". I can not follow this conclusion. Obviously, the association with memory performance first of all suggests that the identified assemblies play a role in the representation of words, most likely on the category level. I can not recognize clear arguments in the discussion refuting this, from my view, most obvious interpretation.
- Page 8: "This resulted in a total of 307 neurons". What was the proportion of single- vs. multi-units?
- Page 8: "We used Morelet wavelet convolution". What was the number of cycles?
- Page 9: "... the binned firing rates". In my opinion, it is incorrect to speak of firing rates, because of the shortness of the time intervals. The authors should rather refer to spike counts.
- Page 9: "... we constructed a neuron-by-time bin matrix." From my understanding, the matrix will to a large part comprise only two different values: 0 Hz and 40 Hz (i.e. absent vs. present spike). The authors should comment on whether their analyses and results are robust against this fact.
- Page 10: "That assemblies were not more localized than expected by chance ...". In my opinion, some degree of localization may actually be expected for these assemblies, for instance, to a certain hemisphere and region. Please test the spatial clustering separately for hemisphere, region and channel.
- Page 12: "... reactivation-triggered power: 40 Hz and 100 Hz." Please add a reference to Supplementary Fig. 5A.
- Page 15: „Umbach, G. et al.“. Please add the publication year.
- Supplementary Table 1: "Epilepsy zone". This is not a clinically appropriate term. Do you mean epileptogenic zone?

Reviewer #1 (Remarks to the Author):

In the paper entitled “Flexibility of functional assemblies supports human memory”, the authors used data from refractory epileptic patients to investigate the role of gamma activity in different brain structures during memory task. The paper is very interesting. Main novelty resides in using advanced methods for unit activity inherited from animal research in human subjects performing a memory task. I find the paper promising and well written in terms of general narrative, which flows logically from premises to conclusions. Method section however, lacks some important details to make it more understandable and, more importantly, reproducible.

We appreciate the reviewer’s concerns regarding the presentation of our methods. We have revised the manuscript in response (e.g., page 3 and 8). Please see below for the specific changes implemented.

For instance, the incredible number of different approaches used to identify and characterize neuronal assemblies would benefit from a graphical representation linking raw data to raster to assemblies, and finally to statistics.

Please see new Supplementary Fig. 2.

Moreover, figures are very blurred in my document which makes very difficult to read some important details such axis information. Things get worst if one tries to zoom in to look at details (e.g. axis labels). That’s annoying also because they all look very dense of well-presented results but difficult to carefully revise. I have several concerns listed below:

We apologize for the poor pixel density in several of the figures initially submitted. We have remedied this for the resubmission.

Data description is very confusing across the entire MS. Numbers keep changing across different paragraphs without clear connection with the unique independent variable that is meaningful which are the different N-subjects. For instance, authors says that the initially enrolled N=26 epileptic patient for a total of 38 recording sessions. How many recording sessions per patients have been performed and which was the criteria for performing in a subset of patients more than one session? Later in the MS, authors states “We isolated a total of 45 gamma timescale assemblies in 15 of the recording sessions” from how many different subjects?

Thank you for pointing this out. We recorded a total of 38 recording sessions from 26 patients with epilepsy, with 17 of the 26 contributing a single session. Our ability to record multiple sessions was dictated by the duration of clinical seizure monitoring. From the 38 recording sessions, we identified 45 neuronal assemblies in a subset of 15 recording sessions. These 15 sessions were from 13 unique subjects with 2 subjects

contributing 2 recording sessions (Supplemental Table 1). Failure to identify assemblies was likely due to low unit yields in these sessions which precludes their detection (see Supplementary Fig. 1J). We invariably identified assemblies in recording sessions during which we isolated more than 12 neurons. We have clarified this in the methods section (page 8) and throughout the manuscript (page 3).

In respect to the task, subjects should recall as many words as they can from a list presented during encoding and after a distractor task (math). The authors said that those are non-repeating words, but are those words also non-semantically related and do the synonyms have been removed? If not, does the assembly characteristics predict encoding of semantically related words (or synonyms) better/worst?

The reviewer raises a reasonable concern regarding the semantic relatedness of the memory items. Memory items were selected at random from a pool of approximately 1000 common nouns without replacement. These words were not intentionally semantically related and there are no synonyms in the dataset. However, we understand the reviewer's concern regarding the impact of semantic similarity on assembly activity. As such, to directly test this question, we calculated a semantic similarity score of each word in the dataset to a reference word using latent semantic analysis as in a previous work (Umbach et al. PNAS 2020). We then grouped the words into 10 different categories based on their similarities to the reference word such that words within bins were more similar to each other than words across bins. We then looked for a relationship between assembly expression strength and the semantic category. We included an expanded explanation of these methods on page 11. If a significant relationship was found, we called that assembly "semantically modulated." Of the 45 assemblies, only 3 were semantically modulated. As such, we do not believe our results regarding assembly reactivation are explained by semantic similarity among recalled items. However, the author's point about the possibility of assemblies responding to semantic information is insightful, and we have added a brief description to the Discussion on page 7 describing future efforts at differentiating drift rates among temporally vs semantically sensitive assemblies utilizing experimental paradigms better suited to such distinctions (Howard and Kahana Journal of Memory and Language 2002).

There is no discussion about the potential role of pathology and/or about AEDs. According to data description, these patients are affected by refractory epilepsy undergoing presurgical evaluation for surgical resection. As far as I know, hippocampal sclerosis is a typical structural malformation in MTL epilepsy. The authors rejected pathological activity (although description of this particular step remains a little confusing, see below comment). Finally, anti-epileptic drugs are known to have mid to large effects on cognitive performances including memory and motor action. These effects are strongly depending on the type of AED administered and duration of treatment. The authors should at least comment the potential role in performance linked to pathology and AEDs.

The reviewer raises a reasonable point regarding some of the inevitable challenges in conducting research in this patient population. We have clarified the methods we employed to reduce the impact of interictal activity which follow those employed in similar publications in humans. While hippocampal sclerosis is commonly found in patients with temporal lobe epilepsy, such patients mostly do not undergo intracranial monitoring (because the lesion identifies seizure onset location). However, microscopic changes consistent with MTS may still be present in our patient population. The impact of the identity and dosages of anti-epileptic drugs is difficult to assess for patients undergoing intracranial monitoring because these are often being adjusted during the hospital admission, and there is substantial heterogeneity in terms of the drug classes and dosages across subjects. We have added a mention of these caveats on page 8.

Method section, line 30: authors state that they flagged and discarded channels with inter-ictal activity while in the following sentence saying that they did not. That's a contradiction that should be fixed.

Thank you for pointing this out. We have clarified this (page 8). Channels with inter-ictal activity were flagged but not discarded. We ensured that we could still identify assemblies without including these channels in the analysis.

Minor:

Fig1 what the numbers next to label mean - e.g. 156 - AH?

This indicates the number of single units isolated from that region. We have clarified in the figure legend (page 17).

Reviewer #2 (Remarks to the Author):

Umbach et al. studied hippocampal neural firing pattern in relation to gamma phase in a free recall task, using human cell data. They showed 25 ms assemblies reflect consistent sequences within gamma oscillations and phase locking. The data is rare and interesting.

But the function wrt. gamma oscillation remains elusive. For example, what is the function of phase locking, does it relate to autoassociative dynamics? e.g., Pfeiffer, B. E., & Foster, D. J. (2015). Autoassociative dynamics in the generation of sequences of hippocampal place cells. Science, 349(6244), 180-183.

The reviewer raises an insightful question regarding the temporal dynamics of assembly activation and their relationship to gamma oscillations. Certainly, the short time scales that characterize gamma-scale assemblies (25 msec) are different than the sequences of activation observed within a 300 msec slow wave ripple or the sequences observed relative to theta oscillations for spatial information. As such, information represented in the temporal patterns likely occurs among rather than within individual assemblies. The reviewer's insight then is to ask what the function of organization according to gamma phase would be, noting that it might provide a mechanism for autoassociation. We

appreciate this suggestion, as we think connecting our findings related to gamma organization of assembly activity to autoassociation helps clarify the importance of this observation. We have added reference to this idea in the Discussion on page 6 with appropriate references.

Similarly, the function of “drift” remains unclear. Throughout paper, there are many metrics being computed but without clear reference to function.

We appreciate the reviewer’s concerns related to assembly drift, which is a relatively novel concept. We define drift as a neuron’s gradual increased or decreased association with the assembly across the experiment. This is measured by correlating the firing rate of individual neurons within moments of assembly activation (as defined in the methods on page 13) with the experimental time. Also, we now state explicitly on page 6 what the possible function of “drift” may be, drawing on studies of rodent assemblies (Hainmueller & Bartos Nature 2018) and recent theoretical work (Mau et al. eLife 2020). We propose that drift may provide a mechanism for updating of memories while retaining core features of their representation, although further investigation will require more explicit experimental manipulation.

Also, I don’t think this the first time neural spiking in temporal lobe is shown to be related to episodic memory. “Neural assemblies” are also more like a analysis product (especially given there are only 13, simultaneously recorded neurons, on average). It is not clear why this is definitively better or revealing things fundamentally different compared to previous findings, e.g., Vaz, A. P., Inati, S. K., Brunel, N., & Zaghoul, K. A. (2019). Coupled ripple oscillations between the medial temporal lobe and neocortex retrieve human memory. Science, 363(6430), 975-978.

We agree with the reviewer that this is not the first time unit activity in humans has been correlated with memory behavior and did not intend to project this impression. It is, to our knowledge, the first study using analytic methods with the explicit intent of defining groups of neurons based on their co-firing activity on the timescale of gamma oscillations, a fundamental concept in theoretical neuroscience (Buzsaki Neuron 2010, Lisman & Idiart Science 1995). Several human single unit studies have defined groups of neurons based on their co-firing across minutes (Gelbard-Sagiv et al. Nature 2008), seconds (Umbach et al. PNAS 2020), or even hundreds of milliseconds (Vaz et al. Science 2020), as we now acknowledge on page 1. We also clarify that our approach incorporating assembly activation is complementary to these previous findings and that further work is needed to connect our work with these efforts (page 6).

There are also some seems-too-strong inferences in the discussion with little empirical evidence, e.g., on page 6, line 210-212 “However, participants with drifting assemblies demonstrated superior memory performance, supporting recent suggestions that dynamic assemblies may facilitate memory by allowing for flexible updating of experiential detail”, I don’t see a clear logic here.

We understand the reviewer's point, and in response we have altered the language used in the manuscript to make clear what our findings related to drift directly demonstrate as compared to previous theoretical treatments and how we interpret our findings in light of these (page 6). We now state what follows:

While further work is needed to elucidate the mechanism and meaning of drift in memory, our finding is consistent with recent suggestions that dynamic assemblies may facilitate memory by allowing for flexible updating of experiential detail¹⁵. It is possible that the drift we observed, particularly in the posterior hippocampus, reflects our ability to update details of experiential context, increasing our ability to orthogonalize events within a broader experience, facilitating memory.

Minor:

Page 4, line 126-130, the author stated "This effect remained after including assembly firing rate preferences for successful encoding, ... , indicating that assembly firing order carries mnemonic information above assembly firing rate." Do authors meant to write "excluding"?

Yes, we meant excluding the effect by including firing rate preferences in the model. This has been clarified (page 4).

Reviewer #3 (Remarks to the Author):

The authors investigated single-unit recordings in epilepsy patients performing a word memorization task with free recall. They describe the contribution of co-firing groups of neurons and local field potentials in the medial temporal lobe to memory formation. Their findings suggest that neuronal assemblies exhibiting correlated firing on a 25 ms timescale and spike-field coupling in the gamma range play an important role in successful long-term memory.

This investigation is based on a precious data set, the analytic approaches appear to be inventive and sound, and the results are interesting and relevant. In some instances, the interpretation of the results is hard to follow, from my view. Overall, I found that the study is analytically and mathematically very strong, but that there are several conceptual weaknesses and confusions. Moreover, from my view important information is missing in the manuscript. Please find enclosed more detailed comments and suggestion. Major comments are marked with underlined page numbers.

- Page 2: "Our study included 26 participants ...". Please add here how many subjects/sessions entered the main analyses.*

We have clarified this in the manuscript (page 3) and this information can be found in the methods (page 8).

- Page 3: "To achieve this, we first binned ... into 25 ms time bins". Please mention that the time bins were non-overlapping.*

We have clarified this in the manuscript (page 3) and the methods (page 9).

- *Page 3: “We isolated a total of 45 gamma timescale assemblies ...”. Please add the information here that this procedure was based on data from the whole recordings sessions.*

We have clarified this in the manuscript (page 3). This information can be found in the last sentence of the first paragraph of the “Assembly identification” section of the methods (page 9).

- *Page 3: “We isolated a total of 203 neurons ...”. Did you differentiate between single- and multi-units?*

The automated spike sorting software we utilized (Niediek et al. PLoS One 2016) makes efforts to separate unique waveforms recorded on the same microwire channel. While we cannot exclude the presence of some multi-units, we calculated the isolation distance between each putative single unit and all other putative single units isolated on the same microwire. The resulting distribution of isolation distances across our dataset is comparable to other high quality single unit datasets (Faraut et al. Scientific Data 2018) (Supplementary Figure 1F).

- *Page 3: “... quality metrics which closely match those in previous studies of the human MTL18”. Please also cite studies from other groups.*

We have cited additional groups who reported their single unit quality metrics (Faraut et al. Scientific Data 2018).

- *Page 3: “We defined assembly reactivation events ...”. The term “reactivation” is misleading in this context. This term would only fit when assemblies were identified in an initial time interval and would be active in a later interval. The authors should, for instance, just speak of assembly activation, as well as of a subsequent memory effect.*

We understand the reviewer’s point in this regard, and we agree that the term “reactivation” may be confusing to readers. As such, we have changed the term “reactivation” to “activation” throughout the manuscript.

- *Page 3: “... we isolated all spiking activity of member neurons ...”. From my view, also the term “member neuron” is potentially confusing. Based on assembly identification, it would be intelligible to call the total of 203 neurons “member neurons”. The subset of neurons with increased firing rates during activation events could, for instance, be called “responsive” assembly neurons.*

We chose this language to fit with the precedent set in previous reports that utilize our PCA/ICA framework for assembly identification (van de Ven et al. Neuron 2016). While we believe “responsive” is a reasonable choice, in the same way each neuron is a

member of the assembly to a variable degree, each neuron is responsive within the assembly activations to a variable degree. We therefore think that for the purpose of cross-study comparison, using the precedent language is preferable.

- *Page 4: "... along with concomitant in power at 100 Hz ...": please elaborate on this finding in the discussion.*

We have added further discussion around the 100 Hz power increase (page 6).

- *Page 5: "... assemblies with higher drift fraction correlated with greater memory recall"; page 6: "... participants with drifting assemblies ...". In my opinion, it is unclear whether these findings indeed support the concept of drifting assemblies. Previous single-unit findings suggest that different words are represented by different groups of neurons, which however may overlap. Due to the overlap neurons can be identified which show a category-related response to visually presented words. These word-responsive neurons may contribute to the responsive assembly neurons identified by the authors. I would suggest that the authors check this by identifying word-responsive neurons and testing the overlap with the responsive assembly neurons. Importantly, the observed drift, in my opinion, may be just a reflection of the fact that different, however overlapping, assemblies represent different words. Hence, I further suggest that the authors reconsider their interpretation of these findings.*

The reviewer raises an interesting point regarding a possible explanation for our observation of drift, which was also echoed by reviewer 1. To test this directly, we calculated a semantic score for each word using latent semantic analysis and comparing each word presented to a reference word in the dataset. Next, we defined "semantically modulated" neurons via the same methods outlined response to R1, except this time running the analysis for each neuron rather than assembly. We found 4 of 85 assembly member neurons and 7 of 119 non-member neurons responded to the meaning of the presented words ($\chi^2 = 0.13$, $p = 0.71$, chi-square test). Without semantically modulated neurons, we think it less likely that the identified drift can be explained by the semantic meaning of the word stimuli. However, we have included these results on page 3 and added further discussion on the concept of drift on page 6, in which we acknowledge this as a possibility that may be investigated in future experiments that incorporate semantically related vs unrelated words as an explicit manipulation.

- *Page 6: "The assemblies we report exhibit clear associations with memory performance ...". Therefore, it is possible that these assemblies represent the mnemonically relevant contextual features of the overall experimental experience ...". I can not follow this conclusion. Obviously, the association with memory performance first of all suggests that the identified assemblies play a role in the representation of words, most likely on the category level. I can not recognize clear arguments in the discussion refuting this, from my view, most obvious interpretation.*

We have both added the aforementioned analyses (page 3) and expounded upon the possible interpretations of assembly meaning (page 6-7) to address this concern. With these new analyses (and appropriate caveats as suggested), we believe the potential relationship between assembly activity and memory context should be clearer to readers.

• *Page 8: “This resulted in a total of 307 neurons”. What was the proportion of single- vs. multi-units?*

These were all putative single units. Please see the earlier response on this topic.

• *Page 8: “We used Morelt wavelet convolution”. What was the number of cycles?*

We used 6 cycles. This has been added to the methods section (page 9).

• *Page 9: “... the binned firing rates”. In my opinion, it is incorrect to speak of firing rates, because of the shortness of the time intervals. The authors should rather refer to spike counts.*

We chose to use this terminology for consistency with previous reports utilizing this method (Van de ven et al. Neuron 2016). However, we have ensured that our results are robust to binning firing rates with our shuffle procedure explained under the subsequent comment.

• *Page 9: “... we constructed a neuron-by-time bin matrix.” From my understanding, the matrix will to a large part comprise only two different values: 0 Hz and 40 Hz (i.e. absent vs. present spike). The authors should comment on whether their analyses and results are robust against this fact.*

We generally agree with this observation, though there is more spread in observed values as neurons frequently burst within a 25 ms timeframe. However, this is the reason we added additional controls to ensure that the identification of assemblies was not due to high z-scored firing rates resulting from using short duration time bins. Firstly, the selected PCA/ICA method is prone to falsely identifying assemblies when the distribution of values used (in our case, the distribution of firing rates within our neuron by time bin matrix) is heavy tailed (Biroli et al. EPL 2007). To limit this, we only included neurons with baseline firing rates of at least 0.5 Hz, as this curtailed the upper limit of z-scored firing rates observed. Further, we included a shuffle control on our assembly identification step. To do this, we shuffled the spike vectors of each neuron randomly before constructing the firing rate matrix. Thus, the distribution of z-scored firing rates was the same in the real and shuffled data, but the relationship of spiking across neurons was permuted. We never identified as many assemblies in the shuffled data as in the real data (Figure 1G), suggesting that our identification of assemblies is not adequately explained by skewed firing rate distributions within the matrices fed into our PCA/ICA framework.

• Page 10: *“That assemblies were not more localized than expected by chance ...”. In my opinion, some degree of localization may actually be expected for these assemblies, for instance, to a certain hemisphere and region. Please test the spatial clustering separately for hemisphere, region and channel.*

We additionally tested for spatial clustering separately for hemisphere, region, and channel. We did so by counting the instances in which member neuron pairs had the same hemisphere, region, and channel. We then counted the number of times each neuron pair from all included recording sessions overlapped by hemisphere, region, and channel. We then performed a chi-square test for each variable separately to see if assembly membership modulated spatial overlap. Results are found in the figure legend of Supplementary Fig. 1. In brief, we noted a marginal but significant spatial clustering of assembly neurons by hemisphere (89% versus 78%, $\chi^2 = 9.4$, $p = 0.0023$), region (56% versus 39%, $\chi^2 = 14.6$, $p = 1.3e-4$), and channel (26% versus 13%, $\chi^2 = 18.0$, $p = 2.2e-5$).

• Page 12: *“... reactivation-triggered power: 40 Hz and 100 Hz.” Please add a reference to Supplementary Fig. 5A.*

We now refer to Supplementary Fig. 5A.

• Page 15: *„Umbach, G. et al.“. Please add the publication year.*

We have added the publication year (2020) (page 15).

• *Supplementary Table 1: “Epilepsy zone”. This is not a clinically appropriate term. Do you mean epileptogenic zone?*

This has been corrected.

REVIEWER COMMENTS

Reviewer #1 (Remarks to the Author):

I'd like to thank the authors for their valuable work and for addressing all my concerns. I don't have any further major comments and I now endorse the publication of the paper in Nature Comm.

I only report a minor error, that can be addressed directly during proof reading. At line 43, the authors write "25 msec" which should read 25 ms, instead.

In conclusion, I'd like also to congratulate with the authors for their interesting paper.

Reviewer #2 (Remarks to the Author):

The paper is improved with better illustrations and more clarifications. But I am afraid the authors have not answered the most important question I have raised. More in-depth analyses should be done.

1. what is special about this gamma phase coupling, e.g., what its relationship to theta (e.g., does the assemblies happen at the trough of theta? and/or if there is a relationship between gamma phase and theta), and what this is related to well-documented sharp-wave ripples during memory retrieval and consolidation.
2. On figure S6A, there are two peaks in the temporal frequency analysis, centred at 40 and 100 Hz. Is the latter one related to sharp-wave ripples? what is the relationship of current findings related to this?
3. I also don't understand the authors' comments to my first question regarding gamma oscillation. The authors stated "Certainly, the short time scales that characterize gamma-scale assemblies (25 msec) are different than the sequences of activation observed within a 300 msec slow wave ripple or the sequences observed relative to theta oscillations for spatial information. As such, the information represented in the temporal patterns likely occurs among rather than within individual assemblies."

what is "slow wave ripples"? Do the authors intend to mean "sharp wave ripples"? If so, a typical sharp-wave ripple is within 200 ms, rather than 300 ms.

I also don't understand why the authors think the information represented in the temporal patterns present among rather than within assemblies? Because it is too slow?

Reviewer #2 (Remarks to the Author):

The paper is improved with better illustrations and more clarifications. But I am afraid the authors have not answered the most important question I have raised. More in-depth analyses should be done.

1. what is special about this gamma phase coupling, e.g., what its relationship to theta (e.g., does the assemblies happen at the trough of theta? and/or if there is a relationship between gamma phase and theta)

We agree with the reviewer that this is an interesting and important question. In response, we looked at theta phase locking properties of assembly activations themselves.

To do so, we first generated a list of all activation times for each assembly across their respective sessions. Next, we extracted the theta phase for each of 3 linearly spaced theta frequencies (3, 5.5, and 8 Hz) at each assembly activation time. We then assessed for phase locking with a Rayleigh test for non-uniformity with a threshold of $p < 0.05$. We performed this calculation for each assembly-channel pair, meaning that we assessed the phase locking of each assembly's activation events to each recording channel's theta within a given recording session. We then FDR-corrected across frequencies and channels with a threshold of $Q = 0.05$. Any assembly demonstrating significant phase locking to at least one frequency on at least one channel after statistical correction was said to be theta-phase locked. A total of 30 of the 45 assemblies demonstrated phase locking ($p < 0.001$, binomial test).

We next tested for the phase to which assemblies locked. We measured the mean phase at which phase locking was found for the frequency with maximum non-uniformity of phase (for the three center frequencies described above.) We found that assembly activation events significantly phase locked near the trough of the theta oscillation for each of the three frequencies used in our analysis. 3 Hz ($z(22) = 6.3$, $p = 0.0014$, 95% confidence interval (CI): 135-199 degrees), 5.5 Hz ($z(25) = 8.5$, $p < 0.001$, 95% CI: 134-186 degrees), and 8 Hz ($z(24) = 8.7$, $p < 0.001$, 95% CI: 103-154 degrees). These data are now displayed in supplemental figure 7 and mentioned in the manuscript on page 5.

We appreciate the reviewer's suggestion in this regard, because our results suggest a relationship between theta and gamma oscillations by which gamma oscillations organize the *timing* of neuronal spiking within an assembly while theta oscillations organize when the assembly fires, at least for a subset of assemblies. The differences among assemblies locking to different theta frequencies remains an important topic for future investigation, and we note the complementary properties of distinct oscillations at distinct frequencies across a broad 2–10 Hz band is a key question in human neuroscience.

...and what this is related to well-documented sharp-wave ripples during memory retrieval and consolidation.

In response to this we performed a ripple analyses following previously established methods (Vaz et al. Science 2019). For each channel within each recording session, we identified ripples of at least 25 ms duration using the aforementioned protocol. This resulted in an average ripple frequency of 0.29 +/- 0.12 Hz across sessions, similar to previous reports in humans. We then looked at the coincidence of ripples and assembly activation bins, finding an average coincidence of 3.7%, which was higher than chance based on a shuffle analysis ($p = 0.005$, 200 shuffles), and reflective of an average relative risk of 3.8, meaning that assemblies were 3.8 times more likely to be active during ripple events that outside of them. These data are now displayed in supplemental figure 7.

2. On figure S6A, there are two peaks in the temporal frequency analysis, centred at 40 and 100 Hz. Is the latter one related to sharp-wave ripples? what is the relationship of current findings related to this?

Based on the above analysis, we do believe that it is possible that the power peak centered at 100 Hz in the mentioned plot is related to sharp-wave ripples. However, we caution that ripples in human data are considerably rare compared to rodent data and further dedicated work is needed to elucidate the recording devices (e.g. Behnke-Fried versus Utah array) and frequency band and prefiltering best suited to discern potential ripple from epileptic activity or physiologic high frequency oscillations. Further, the free recall paradigm imposes limited behavioral structure on item retrieval, which reduces our ability to observe ripple events as compared to spatial navigation paradigms.

3. I also don't understand the authors' comments to my first question regarding gamma oscillation. The authors stated "Certainly, the short time scales that characterize gamma-scale assemblies (25 msec) are different than the sequences of activation observed within a 300 msec slow wave ripple or the sequences observed relative to theta oscillations for spatial information. As such, the information represented in the temporal patterns likely occurs among rather than within individual assemblies." what is "slow wave ripples"? Do the authors intend to mean "sharp wave ripples"? If so, a typical sharp-wave ripple is within 200 ms, rather than 300 ms.

Yes, we intended to use the term sharp wave ripples. Further, we have ensured that we have not included any erroneous estimate of ripple duration throughout the manuscript.

I also don't understand why the authors think the information represented in the temporal patterns present among rather than within assemblies? Because it is too slow?

We made this statement as we defined our assemblies based on coactivation within a single gamma oscillation (~25 ms). Therefore, phenomena of greater duration, such as the sequences observed during replay events or theta precession are more likely comprised of sequential activation of these fundamental units (Buzsáki Neuron 2010).

We have omitted this language from the manuscript in response to reviewer concerns over the lack of clarity.

REVIEWERS' COMMENTS

Reviewer #2 (Remarks to the Author):

The authors have answered all my questions. I would like to congratulate the authors for their valuable work.